# Biomimetic Transparent Eye Protection Inspired by the Carapace of an Ostracod (Crustacea)

**DOI:** 10.3390/nano11030663

**Published:** 2021-03-08

**Authors:** Andrew R. Parker, Barbara P. Palka, Julie Albon, Keith M. Meek, Simon Holden, F. Tegwen Malik

**Affiliations:** 1Green Templeton College, University of Oxford, 43 Woodstock Road, Oxford OX2 6HG, UK; 2School of Optometry and Vision Sciences, Cardiff University, Cardiff CF24 4HQ, UK; Barbara.p.palka@gmail.com (B.P.P.); albonJ@cardiff.ac.uk (J.A.); MeekKM@cardiff.ac.uk (K.M.M.); 3Defence Science and Technology Laboratory, Physical Protection Group, Porton Down, Salisbury SP4 0JQ, UK; sjholden@mail.dstl.gov.uk; 4Bay Campus, School of Management, Swansea University, Swansea SA1 8EN, UK; f.t.malik@swansea.ac.uk

**Keywords:** ostracod, biomimetics, transparency, scratch resistance, impact resistance, armoury

## Abstract

In this study we mimic the unique, transparent protective carapace (shell) of myodocopid ostracods, through which their compound eyes see, to demonstrate that the carapace ultrastructure also provides functions of strength and protection for a relatively thin structure. The bulk ultrastructure of the transparent window in the carapace of the relatively large, pelagic cypridinid (Myodocopida) *Macrocypridina castanea* was mimicked using the thin film deposition of dielectric materials to create a transparent, 15 bi-layer material. This biomimetic material was subjected to the natural forces withstood by the ostracod carapace in situ, including scratching by captured prey and strikes by water-borne particles. The biomimetic material was then tested in terms of its extrinsic (hardness value) and intrinsic (elastic modulus) response to indentation along with its scratch resistance. The performance of the biomimetic material was compared with that of a commonly used, anti-scratch resistant lens and polycarbonate that is typically used in the field of transparent armoury. The biomimetic material showed the best scratch resistant performance, and significantly greater hardness and elastic modulus values. The ability of biomimetic material to revert back to its original form (post loading), along with its scratch resistant qualities, offers potential for biomimetic eye protection coating that could enhance material currently in use.

## 1. Introduction

Myodocopid ostracods (‘seed-shrimps’) are small, marine crustaceans from 1 to 32 mm in length [1]. Their shrimp-like bodies are enclosed within a relatively thin (20–60 µm thick), bi-valved carapace. The two valves of the carapace are connected by a hinge, and can open and close while retaining their rigidity. Many myododcopids are benthic (most up to 500 m depth), burying up to 1cm depth in the sediment (and compressed by sand particles), while some are pelagic, bombarded by suspended particles (such as faecal pellets, carcases, crustacean moults [2] that move through the water at varying speeds. Predatory myodocopids, such as the relatively large, pelagic *Macrocypridina castanea* (Figure 1), may also need to defend themselves against prey, such as the terminal claws of small crustacean limbs, while in the grasp of the ostracod’s elongated mandibles that protrude through the opening in their carapace. Consequently, the carapace must offer protection against relatively large and abrasive sand grains or particles, or crustacean claws, which could cause indentations, cracks or scratches in this protective shell.

Most myodocopids can be distinguished from other ostracod taxa by their well-developed compound eyes, some occupying up to a quarter of their total body size [3]. This study focuses on the myodocopid family Cypridinidae (which has around 30 genera), known to use visual signals in the form of iridescence or bioluminescence for courtship [4,5]. Therefore, the uniqueness of these species is that not only do they have compound eyes but they also possess highly transparent yet protective carapaces, through which they must see [6].

While some species of myodocopids possess transparent windows in their carapace in the regions covering the eyes only, others possess a carapace with uniform transparency [6]. Parker et al. [6] discovered that the transparent properties of the carapace of the cypridinid *Macrocypridina castanea* were aided the carapace’s thin-layered (internal laminar stack) construction (Figure 2). The largest, laminate section of this carapace was found to be composed of a chitin-protein complex and low magnesium calcite with alternating high-low index material layers, each with an optical thickness (actual thickness multiplied by the refractive index) of ½ wavelength of blue light. This half-wave stack neither reflects nor absorbs light in the wavelengths in which cypridinid ostracods see, and therefore provides transparency [6]. In a similar manner, the human transparent cornea not only allows visible light to pass through it unimpeded but has the additional function of protecting the eye by acting as an outer casing [7].

The remaining two, thinner sections of the carapace covering the eye of *Macrocypridina castanea* occupy a total of about one-third of the carapace thickness, and are the outermost sections. These (particularly the very outer layer) possibly function to prevent the initiation of scratching (i.e., preventing an object moving parallel to the shell surface from ‘digging in’) whereas the section containing the thin-layer stack (Figure 2d) probably functions (at least in part) to halt scratch or crack propagation if the outer layers become compromised in some way. Whilst the optical properties of the thin-layer stack have been studied, little is known about its mechanical properties.

Such a transparent, protective eye shield would be relevant to the area of protective eye wear used by the military or police, or in industrial settings. Effective eye armoury must take into consideration multiple mechanical and optical functions such as scratch resistance, adequate indentation performance, prevention of crack propagation and optical transparency. It is the aim of this study to mimic the thin-layer stack of the ostracod carapace and assess its viability as eye wear armour, potentially enhancing solutions already in use.

## 2. Materials and Methods

The carapace structure of the transparent window of the cypridinid *Macrocypridina castanea* informed the fabrication of human-made analogue material and its subsequent testing to assess its mechanical properties. The fabrication and testing methods are set out in this section. For each stack type type (5, 10 and 15 bi-layered stack), one sample was tested twice.

### 2.1. Fabrication of Thin-Layer Stack 

The 5, 10 and 15 bi-layered stacks (of alternate high and low index materials), identical in size and morphology to those of the ostracod carapace’s thin-layer stack, were deposited on glass microscope slides (of refractive index n ≈ 1.5) using vacuum coating machines (according to Parker et al. [6]). These three coatings were each tested to determine the effect of additional layers.

The bi-layers consisted of silicon dioxide (SiO_2_), with a ‘low’ refractive index (n) of 1.46, and zirconium dioxide (ZrO_2_), with a ‘high’ refractive index of 2.16. The coatings were initially tested against an uncoated microscope slide; the best performer was used for the final phase of testing against the appropriate reference materials: transparent polycarbonate (typically utilised in protective military lenses), a scratch resistant coated spectacle lens (i.e., a glass lens dip coated with Poly-siloxane) and an uncoated spectacle glass lens.

### 2.2. Initial Assessment of Biomimetic Eye Protective Material 

The coated samples were tested and compared against the reference materials. Two different methods were employed to assess the fabricated surfaces for their mechanical properties: indentation testing and scratch resistance testing.

### 2.3. Indentation Testing

The initial testing of the 5, 10 and 15 bi-layer coated microscope slides were carried out using an Ultra Nano Indentation Tester (UNHT), Anton Paar, Buchs, Switzerland. This instrument was used to assess the hardness and elastic modulus of the samples. The UNHT was chosen as it enabled a controlled loading that minimised any effect the substrate may have on the measurements. Two types of tests were carried out on the coated samples, namely quasi-static low load testing (to assess the hardness and elastic modulus of the top coating), and sinus mode testing (to analyse the mechanical properties with penetration depth). Both tests were conducted in air at a temperature of 24 °C and humidity 40%, and both used Berkovich indentation tips. Hardness and elastic modulus results were determined using the Oliver and Pharr (1992) method [8].

The linear low load tests had a maximum load of 50 µN (and pausing at this maximum load for 10 seconds) applied to each sample with a loading and unloading rate of 100 µN/min. Several indentations were performed on each of the three coated samples with a maximum penetration depth set to ~20 nm (approximately 10% of the thickness of the coating’s top layer).

The sinus mode tests were used to further analyse each sample’s mechanical properties (hardness and elastic modulus with penetration depth). A constant strain rate was applied with a minimum load of 50 µN and a maximum load of 10 or 40 mN. The sinus frequency was set at 10 Hz and sinus amplitude at 1 mN with two tests carried out per sample, one up to 10 mN and the second up to 40 mN with penetration depths of ~330 nm and 690 nm, respectively.

### 2.4. Scratch Resistance Testing

A Nano Scratch Tester (NST), Anton Paar, Buchs, Switzerland, was used to carry out the initial scratch testing on the fabricated 5, 10 and 15 bi-layer coatings. These tests were carried out in air at a temperature of 24 °C and 40% humidity. A sphero-conical indenter was used with an indenter radius of 10 µm. Progressive loading was applied with the following parameters: scanning load of 4 mN, initial load of 4 mN, final load of 800 mN, loading rate of 1592 mN/min, scratch length of 1 mm carried out at a speed of 2mm/min. The force feedback loop control of the NST along with the pre-scan of the sample surface, ensured that surface topography did not affect the force applied and that the real scratch penetration depth was measured. This also ensured that the elastic recovery during the post-scan procedure could be effectively characterised. Visual identification was also used to assess where the first surface cracks appear (first failure LC1) on the scratch track and where complete failure (LC2) of the coating/substrate occurred.

### 2.5. Final Testing of Protective Biomimetic Eye Shield Material

Having conducted initial testing of samples that mimic the thin-layer stack found in the transparent carapace window, a further round of testing was carried out to assess this structure’s ability to perform against material that is used as protective eye wear. Only the 15 bi-layered sample was tested during this phase, but this time compared with appropriate reference materials that are currently in use, namely polycarbonate (material used in military eye wear; Clear Scratch Resistant Perspex^®^ obtained from Cut Plastic Sheeting, Ivybridge, UK, and commercially available anti-scratch spectacle lenses (namely glass lenses dip coated with Poly-siloxane). An uncoated glass spectacle lens was also tested. Again, scratch resistant tests and indentation tests were carried out.

### 2.6. Indentation Testing

Each sample was tested under the same conditions as for the previous tests (in air, at a temperature of 24 °C and at a humidity of 40%) and test parameters (maximum load of 50 µN (with a pause of 10 s at the max load), loading and unloading rate of 100 µN/min and again under quasi-static conditions) using an Ultra Nano-indentation Tester at Anton Paar.

After the linear loading was performed, the sinus mode testing was carried out but this time with a maximum load of 2 mN being set so as to reach similar penetration depths as the initial sinus mode testing performed on the 15 bi-layer coated sample. A loading rate/load was set at 0.1 s^−1^ with a sinus frequency of 5 Hz and sinus amplitude of 1 mN.

### 2.7. Scratch Testing

Each sample was again tested using the Nano Scratch Tester at Anton Paar and under the same conditions as previously used (i.e., in air, at a temperature of 24 °C and at a humidity of 40%). The same sphero-conical indenter was used, with an indenter radius of 10 µm. Progressive loading was applied with the following parameters: scanning load of 4 mN, initial load of 4 mN, final load of 800 mN, loading rate of 1592 mN/min, scratch length of 1 mm carried out at a speed of 2 mm/min.

## 3. Results and Discussion

Prior to mechanical tests, the manufactured 15 bi-layered material on glass slides was confirmed to be transparent. Light transmission at the normal was recorded at ~80% for wavelengths between ~500–800 nm, dropping sharply to ~50% at a wavelength of 400 nm and 0% at a wavelength of 380 nm. This demonstrates a relatively broadband transparency (some transparency was lost through absorption by the glass substrate, and reflection from the lower, uncoated surface of this glass slide).

### 3.1. Initial Assessment of Coated Bi-Layer Slides

#### 3.1.1. Indentation Testing

As expected, due to the same top coating material, the low load indentation tests did not reveal any major differences between the 5, 10 and 15 bi-layered samples (Figure 3a). However, the 15 bi-layer sample illustrated a marginally greater elastic modulus. Assessing the mechanical properties for both coated and uncoated samples, the substrate (microscope slide) was taken into account because its influence was expected to be more important to the five bi-layered sample (i.e., the thinnest coating). However, even though the uncoated sample was found to have a higher hardness result than the coated samples, the results on the coated samples were all very similar (the elastic modulus of the uncoated sample was found to be lower than the coated samples, indicating that it possesses a lower atomic bonding). The coated samples were found to have higher hardness values for the top layer (~7100–8600 MPa) after which the value dropped to ~6000 MPa which, over the 600 nm depth tested, gradually increased to ~6250 MPa (Figure 3b). However, overall, no major differences could be observed between the three coated samples for hardness or elastic modulus (Figure 3c).

#### 3.1.2. Scratch Resistance Testing

The scratch resistance of the samples was assessed and compared with each other. Two major events were observed during the tests on the three samples: an initial crack (LC1) of the sample followed by the complete failure (LC2) of the coating/substrate system.

Initial cracks (LC1) resulted at loads of 154.9 ± 6.0 mN, 192.8 ± 5.2 mN and 182.4 ± 17.1 mN for the 5, 10 and 15 bi-layered coatings, respectively and complete failure (LC2) resulted at loads of 260.0 ± 9.4 mN, 398.3 ± 11.8 mN and 452.2 ± 42.6 mN for the 5, 10 and 15 bi-layered coatings, respectively (Figure 4). Hence the 15 bi-layered coating demonstrated the greatest scratch resistance, requiring the highest load to reach complete failure.

The optical observations (Figure 5) of the samples indicated a substrate failure during the scratch test (at LC2). It can be seen in Figure 5d that the adhesion of the layers to each other appears to be strong, since the bi-layer coating can be observed to have been peeled away from the substrate only upon reaching complete failure and not earlier. It was uncertain whether this complete failure (LC2) of the substrate was triggered by the failure of the substrate itself or by the failure of the layered coatings.

### 3.2. Mechanical Testing of Fabricated Protective Eye Shield Material

From the initial round of testing, because the 15 bi-layered coated sample performed the best with regards scratch resistance, a 15 bi-layered coated sample was used for further testing to compare with reference materials that are known to perform well in the field of scratch resistance on transparent material. Again, both comparative scratch and indentation testing were carried out but this time on the 15 bi-layered coated sample, polycarbonate sample, anti-scratch spectacle lenses (coated with Poly-siloxane) and an uncoated spectacle lens. Both sides of polycarbonate and dip-coated spectacle lens were tested in case of any differences in the material surfaces due to manufacturing processes.

#### 3.2.1. Indentation Testing

As can be seen in Figure 6, the hardness results obtained show that the hardness value (i.e., the resistance of a material to deformation by surface indentation) for the 15 bi-layered coated sample was significantly higher (3865.0 ± 618.6 MPa) compared to the three other samples (between 175.5 MPa and 348.7 MPa). The same can also be said of the elastic modulus data seen in Table 1; the 15 bi-layered sample had an elastic modulus of 92.5 ± 16.6 GPa compared with between 2.9 GPa and 3.5 GPa for the other samples. This higher elastic modulus of the 15 bi-layered sample indicates that it is a stiffer material compared to the other samples, hence the atomic bond strength is greater, and the material undergoes less strain when under load.

These results indicate that the thin-layered structure can act as an underlying protective layer. That is, it is resistant to permanent deformation such as dents, along with having a high stiffness value due to its atomic bond strength, leading to smaller strains in the material under loading. Hence this thin-layered stack in the carapace is able to revert back to its original form when under typical loading from the environment. This demonstrates that this multilayer-type structure has memory-shape mechanical properties when subjected to forces that could be experienced in the field of military eye wear.

#### 3.2.2. Scratch Testing

As plastic deformation is one of the main considerations when analysing the scratch resistance of a material, this was the key focus when analysing the scratch test results. Hence, having carried out the scratch testing, comparative graphs were drawn up (Figure 6) that illustrate the mean residual depth (Rd) against scratch distance, which represent the permanent plastic deformation measured on the scratch test track from post-scan analysis. As can be seen from these graphs, the 15 bi-layered coated sample had the lowest plastic deformation (and thus the best scratch resistance) under normal applied loads, followed by the lens (uncoated and coated) and then the polycarbonate samples (permanent plastic deformation was observed to be highest for these samples). No significant differences were found between the two sides of the lens (as expected, since they were dip coated) and for both sides of the polycarbonate. Surprisingly, no major differences were observed between the coated and uncoated lens samples.

Further studies could be conducted to test the mechanical properties of the outer two sections of the carapace of *Macrocypridina castanea*, albeit work would be required to enable the manufacture of mimetic versions, since the ostracod carapace itself is too thin and small to be tested with existing machines.

## 4. Conclusions

Having previously analysed the ultrastructure of a transparent ostracod carapace, in this study we presented how to successfully fabricate the carapace’s thin-layer stack to produce a biomimetic 15 bi-layered coated sample using thin film deposition. This fabricated carapace structure was assessed for its ability to withstand the type of forces found in the ostracod’s environment. Indentation tests revealed that this thin-layer stack possesses high hardness and elastic modulus values along with resistance to scratching when compared to typical material that is currently employed to protect the human eye. These results are very promising for the fields of transparent armour and eye protection; the 15 bi-layered coating has potential to add value to current ballistic eye wear in use.

The hardness value for the fabricated thin-layer stack of the carapace (15 bi-layered coating) was found to be between 11× and 22× higher than the other materials tested, which illustrates the thin-layer stack’s resistance to permanent plastic deformation (i.e., scratch resistance). It is known that if a material is hard enough, it will blunt a projectile tip, effectively increasing its ballistic performance, which is a primary requirement of armour [9]. Hence these results are further encouraging for the field of ballistics and protective eye wear. However, maybe the most intriguing finding is the carapace’s ability to reform back to its original shape after deformation, which consequently allows the material to be thinner. This has the potential to be applied where protection is required from indentation by loose material/chippings, such as during military field operations and in engineering workshops.

In terms of ostracod adaptation, *Macrocypridina castanea* has evolved a not only transparent but also mechanically strong carapace, at least in the position of its eye. Mechanically, the carapace can withstand the type of forces expected in its natural habitat, including scratching by the sharp, robust limbs of crustacean prey while in the grip of the ostracod’s mandibles if the outer layer were to be compromised in some way.

## Figures and Tables

**Figure 1 nanomaterials-11-00663-f001:**
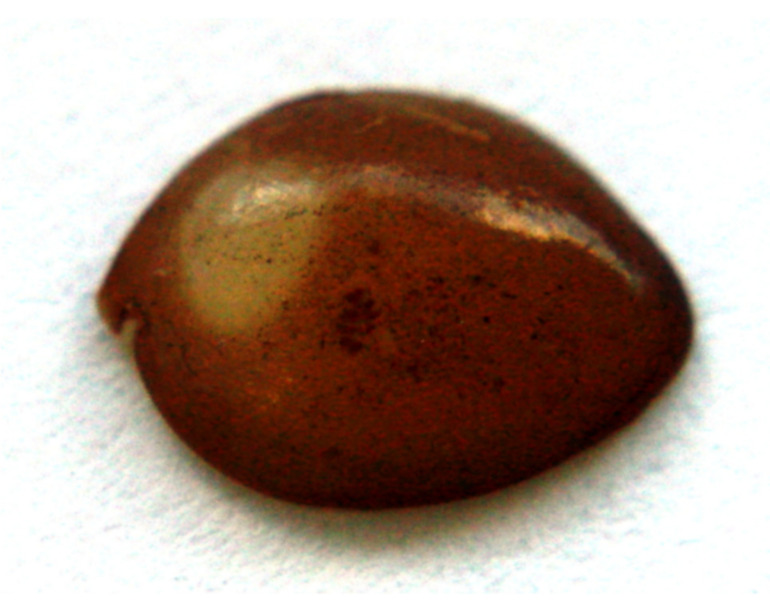
Macrocypridina castanea. Adult male, left carapace, 8 mm long. Anterior to the left. The region that covers the compound eye is visible as a ‘clear’, transparent oval shape (although has been stained by preservation solution). Reproduced from Parker et al. [6], with permission from The Royal Society, 2019.

**Figure 2 nanomaterials-11-00663-f002:**
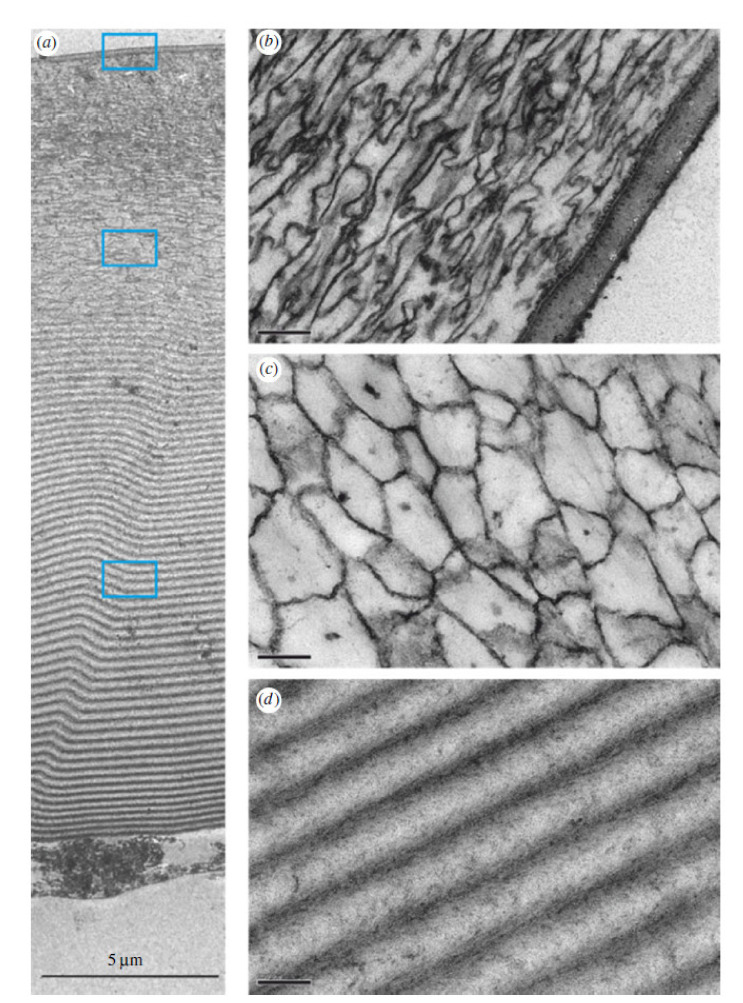
Cross-section of the window of the carapace of *Macrocypridina castanea*, 100 nm thick, stained with uranyl acetate and lead citrate, outermost region at the top, transmission electron micrographs (note a slight distortion in the sample). (**a**) Whole section of the shell. (**b**–**d**) Magnifications of the three selected regions in (**a**), shown by rectangles. Scale bars, 5 µm (**a**), 200 nm (**b**–**d**). Reproduced from Parker et al. [6], with permission from The Royal Society, 2019.

**Figure 3 nanomaterials-11-00663-f003:**
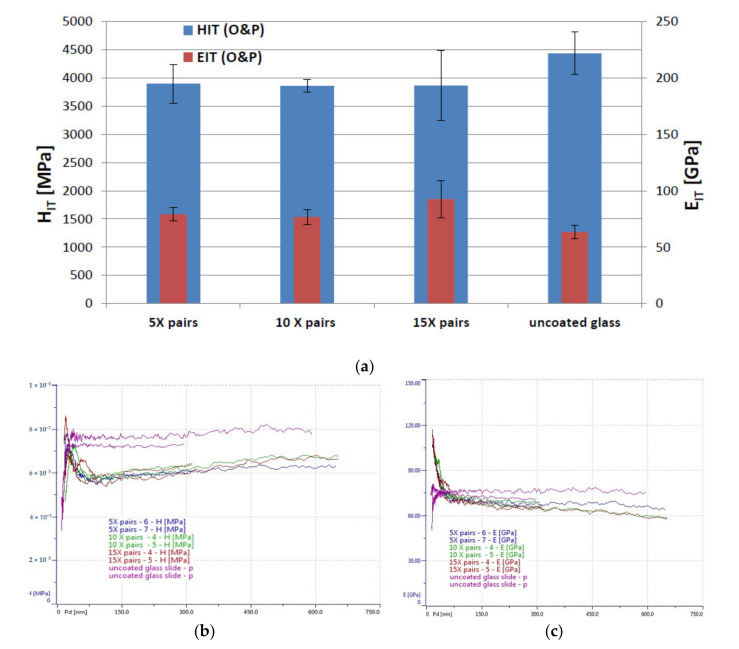
Indentation Testing Results for the 5, 10 and 15 pairs of thin layers on glass, and uncoated glass. (**a**) Comparative graph illustrating the hardness and elastic modulus for the bi-layer coated samples captured during the low load testing. (**b**) Hardness [MPa] evolution with depth curves (2 × 10^−3^ refers to 2000 MPa and 1 × 10^−4^ refers to 10,000 MPa). (**c**) Elastic modulus [GPa] evolution with depth curves.

**Figure 4 nanomaterials-11-00663-f004:**
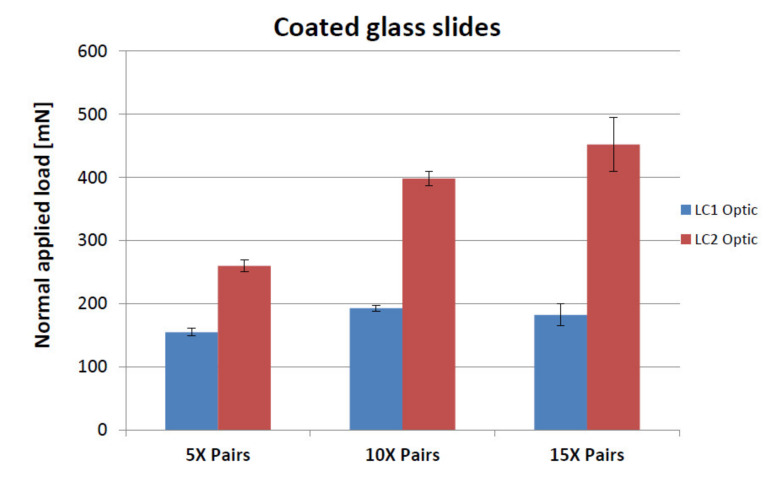
Comparative graph illustrating scratch resistance for the 5, 10 and 15 pairs of thin layers. LC1 is where the first cracks start to appear on the surface test track and LC2 is where there is complete failure of the coating. The 15× bi-layered coating statistically withstood the greater load before complete failure; statistically there was no difference in the load for when the first crack appeared between the 10× and 15× bi-layered stacks.

**Figure 5 nanomaterials-11-00663-f005:**
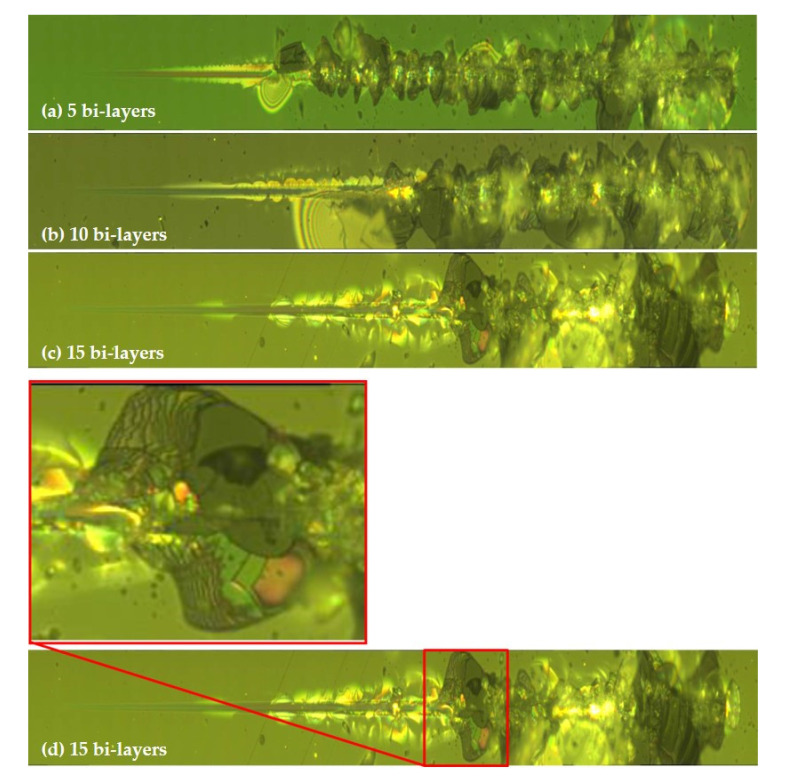
Scratch test track optical images. (**a**) 5 bi-layer scratch track. (**b**) 10 bi-layer scratch track. (**c**) 15 bi-layer scratch track. (**d**) 15-layer test track and zoomed-in region illustrating the clear 15 layers that appear to have peeled away from the substrate.

**Figure 6 nanomaterials-11-00663-f006:**
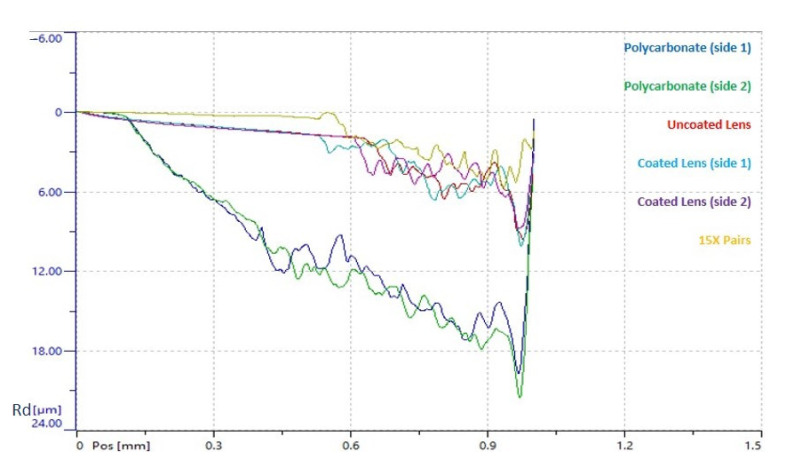
Comparative Curves Mean residual depth curves (permanent plastic deformation) measured after scratching the surface for four samples (15 bi-layered coated sample, uncoated and coated glass lenses, and polycarbonate). ‘Rd’ (y-axis) refers to residual depth of the scratch (µm); ‘Pos’ (x-axis) refers to position or distance from the starting point of the scratch (mm).

**Table 1 nanomaterials-11-00663-t001:** Hardness and Elastic Module Comparative table. Hardness Module (H_IT_ in MPa) data for each sample (both sides of a Polycarbonate sheet (Sample 1), where the marked side Woodward was scratch-resistant-coated, an uncoated lens (Sample 2), both sides of a dip coated lens (Sample 3) and a 15× bi-layered coating) was measured six times and the mean and standard deviation calculated. Likewise, the same approach was taken for the elastic modulus data (E_IT_ in GPa).

Hardness & ElasticModulues	Test Points	Sample 1–PolycarBonate (Marked Film Side)	Sample 1–PolycarBonate (Unmarked Film Side)	Sample 2–Uncoated Lens	Sample 3–Coated Lens (Side 1)	Sample 3–Coated Lens (Side 2)	15× Pairs–Previous Tests
**H_IT_ (O&P) [MPa]**	Data: 1	308.2	362.5	176.7	176.1	177.6	3307.8
	Data: 2	333.4	355.1	195.5	178.0	181.3	3756.7
	Data: 3	315.3	348.8	193.6	177.4	189.4	4530.8
	Data: 4	321.7	354.0	191.3	174.0	181.4	-
	Data: 5	313.9	336.9	183.2	170.0	166.7	-
	Data: 6	338.1	334.7	189.9	177.7	173.1	-
	**Mean**	**321.8**	**348.7**	**188.4**	**175.5**	**178.2**	**3865.1**
	Std Dev	11.7	10.9	7.1	3.1	7.8	618.6
**E_IT_ (O&P) [GPa]**	Data: 1	3.5	3.2	3.2	3.0	3.0	73.5
	Data: 2	3.6	3.2	3.1	3.0	3.0	103.8
	Data: 3	3.5	3.0	3.1	3.0	3.0	100.2
	Data: 4	3.6	3.1	3.1	3.0	2.8	-
	Data: 5	3.4	3.0	3.5	2.9	2.7	-
	Data: 6	3.5	3.1	2.9	3.0	2.9	-
	**Mean**	**3.5**	**3.1**	**3.1**	**3.0**	**2.9**	**92.5**
	Std Dev	0.09	0.06	0.20	0.04	0.09	16.6

## Data Availability

All data collected are presented in this paper.

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
