# Peer review of "Biomimetic Transparent Eye Protection Inspired by the Carapace of an Ostracod (Crustacea)"

_nanomaterials, 2021, doi:10.3390/nano11030663_

Round 1

Reviewer 1 Report

Please see attached pdf.

Reviewer 2 Report

The authors propose a novel eye protection coating based on biomimetic technics. Their initial observation on the crustaceans is sound, the materials and methods are properly documented, results and discussions are all given in a well written presentation. I recommend the manuscript to be published, however I ask the authors for taking into consideration the following minor issues.

  • Title: The version "Biomimetic transparent eye protection inspired by the carapace of an ostracod (Crustacea)" sounds me better.
  • The whole organism crustacean Macrocypridina castanea should be figured, and in the figure the eye inspired the biomimetic study should be pointed by an arrow. That makes the reader to imagine more easily what is going on;
  • Figure 1. The legend is insufficient. The blocks a-b-c-d should be explained in the legend and referred properly in the text;
  • Line 82. A brief but exact description of the subject species is necessary (specific characters, where it lives concretely, etc.), vouchers and depositors of the material examined should be indicated; this is necessary for the repeatability of the experiment (as another species should give slightly different results);
  • Line 127. Previously in line 106 it was written "Anton Paar, Switzerland", here now "Anton Paar (Switzerland)"; should be indicated in the same manner;
  • Line 150. It is disturbing to refer to the previous tests and to give the same data; it is an unnecessary repetition;
  • Line 189. There is no Figure 2";
  • Line 190. The indication (c) should not be inside the panel of (b);
  • Line 218. Figure 5 should be better composed; the inset of "(d)" seems to be deformed (which one is the original?);
  • Lines 274-277 would better fit for the end of the Conclusions;
  • Line 280. Instead of "This study" would be better to write "in this study we presented how to fabricate successfully" or something similar;
  • Line 309. The sections Acknowledgements is not properly written;
  • Line 312: The section CI is not properly written.

Author Response

Changes made to the manuscript following reviewer’s comments

Andrew Parker, 25 February 2021

Reviewer 2

I would like to thank this second reviewer also for their very careful and helpful review. In all cases their comments have led to improvements in the manuscript – all the comments have been addressed and led to changes, as outlined below (the reviewer’s comments are in blue). Please note that the changes have been made on the manuscript using Track Changes.

  • Title: The version "Biomimetic transparent eye protection inspired by the carapace of an ostracod (Crustacea)" sounds me better.

This change has been made.

  • The whole organism crustacean Macrocypridina castanea should be figured, and in the figure the eye inspired the biomimetic study should be pointed by an arrow. That makes the reader to imagine more easily what is going on;

This change has been made. An image of this species has been included as a new Figure 1. The original Figure 1 has become Figure 2, which also solves the problem of the missing Figure 2.

  • Figure 1. The legend is insufficient. The blocks a-b-c-d should be explained in the legend and referred properly in the text;

This change has been made.

  • Line 82. A brief but exact description of the subject species is necessary (specific characters, where it lives concretely, etc.), vouchers and depositors of the material examined should be indicated; this is necessary for the repeatability of the experiment (as another species should give slightly different results);

This change has been made.

  • Line 127. Previously in line 106 it was written "Anton Paar, Switzerland", here now "Anton Paar (Switzerland)"; should be indicated in the same manner;

I have now used “Anton Paar (Switzerland)” throughout.

  • Line 150. It is disturbing to refer to the previous tests and to give the same data; it is an unnecessary repetition;

I would be happy to delete this repetition, but it is listed to confirm exactly which parts we are referring to.

  • Line 189. There is no Figure 2";

This is now resolved (see above)

  • Line 190. The indication (c) should not be inside the panel of (b);

This problem has been solved (“(c)” has been moved).

  • Line 218. Figure 5 should be better composed; the inset of "(d)" seems to be deformed (which one is the original?);

I’m afraid I cannot format this figure. I agree that part (d), and it’s insert, of Figure 5 should me indented so that the left margins are in line with the left margins of (a-c) above it. Can you possibly make this formatting change, please? Thank you.

  • Lines 274-277 would better fit for the end of the Conclusions;

This has now been moved to the Conclusions.

  • Line 280. Instead of "This study" would be better to write "in this study we presented how to fabricate successfully" or something similar;

This change has been made.

  • Line 309. The sections Acknowledgements is not properly written;

This has been changed.

  • Line 312: The section CI is not properly written.

This has been changed.
